# Key Maternity Care Stakeholders’ Views on Midwives’ Professional Autonomy

**DOI:** 10.3390/healthcare11091231

**Published:** 2023-04-26

**Authors:** Joeri Vermeulen, Ronald Buyl, Ans Luyben, Valerie Fleming, Maaike Fobelets

**Affiliations:** 1Department Health Care, Brussels Centre for Healthcare Innovation, Erasmus Brussels University of Applied Sciences and Arts, 1090 Brussels, Belgium; 2Faculty of Medicine and Pharmacy, Department of Public Health, Biostatistics and Medical Informatics Research Group, Vrije Universiteit Brussel (VUB), 1090 Brussels, Belgium; 3Centre for Midwifery, Maternal & Perinatal Health, Bournemouth University, Bournemouth BH1 3LH, UK; 4Frauenzentrum (Centre for Women’s Health), Lindenhofgruppe, 3012 Bern, Switzerland; 5Faculty of Health, Liverpool John Moores University, Liverpool L3 5UX, UK; 6Department of Teacher Education, Vrije Universiteit Brussel (VUB), 1040 Brussels, Belgium

**Keywords:** midwives, midwifery, midwifery autonomy, autonomy, professional autonomy, professionalisation

## Abstract

Advancement towards the professionalism of midwifery is closely linked to midwives’ professional autonomy. Although the perspectives of Belgian midwives on their professional autonomy have been studied, the views of other maternity care stakeholders are a blind spot. The aim of this study, therefore, was to explore maternity care stakeholders’ views on Belgian midwives’ professional autonomy. A qualitative exploratory study was performed using focus group interviews. A heterogenous group of 27 maternity care stakeholders participated. The variation between midwives, with different levels of autonomy, was reported. The analysis of the data resulted in five themes: (1) The autonomous midwife is adequately educated and committed to continuous professional further education, (2) The autonomous midwife is competent, (3) The autonomous midwife is experienced, (4) The autonomous midwife assures safe and qualitative care, and (5) The autonomous midwife collaborates with all stakeholders in maternity care. A maternity collaborative framework, where all maternity care professionals respect each other’s competences and autonomy, is crucial for providing safe and quality care. To achieve this, it is recommended to implement interprofessional education to establish strong foundations for interprofessional collaboration. Additionally, a regulatory body with supervisory powers can help ensure safe and quality care, while also supporting midwives’ professional autonomy and professionalisation.

## 1. Introduction

Professional autonomy, a cornerstone of midwifery’s philosophy [1], is considered a catalyst for advancing midwifery’s journey towards professionalism [2]. In some, but not all, countries midwives’ professional autonomy is limited, which contrasts with the (1) legal framework, (2) evidence about the positive outcomes of midwife led care, and (3) international calls for strengthening midwifery [3]. Midwives may not be able to practice to their full extent in current maternity care settings [4]. Internationally, the medicalisation of birth is suggested as a limiting factor in midwifery autonomy [5,6]. The historical value given to specialist medical services has an impact on midwives’ autonomy, for example, obstetricians mostly perform births in Belgium. Internationally, hospital-based midwives have limited control over the organisation of their work, such as one-to-one care, continuity of care, or working hours, as this is mostly determined by hospital management [7].

Belgium is a country in Western Europe with a complex political organization which is structured on both regional and linguistic grounds. It is divided into three highly autonomous communities and regions: Flemish Region (Dutch-speaking), Walloon Region (French-speaking), and the Brussels Capital Region (bilingual) [7]. In 2019, 10,501 midwives were professionally active in Belgium, 70% (n = 7357) in Flanders, 23% (n = 2400) in Walloon, and 6% (n = 625) in the Brussels Capital Region. Of the total of all professionally active midwives, 78% (n = 8243) work in a hospital setting, 9% (n = 990) in primary care, and 12% (n = 1268) combine both [8]. Maternity services in Belgium occupy an important place in the hospital landscape, and they play a vital role given that about 98% of births in Belgium take place in hospitals [9,10]. Giving birth in a birth centre or at home is rather exceptional. Homebirths are, in general, supervised by independent midwives working in primary care settings. To our knowledge, there are 13 birth centres and one midwifery-led birth centre in Belgium [7]. The effective figures for midwifery-led care in labour and childbirth is uncertain. The second annual report on midwife-led care in Belgium notes that 1.7% of the total number of births in 2021 were midwifery-led [11]. However, underreporting is suspected as the data are from self-registration and Walloon’s data are partly missing.

The national birth rate in Belgium in 2021 was 117.914 [12], with a perinatal mortality rate of 6.0‰ in 2015 [13] and a maternal mortality rate of 6.2 deaths per 100.000 live births in 2017 [12]. Intervention rates in the different regions in Belgium are comparable: induction of labor 27.0%, 30.7%, and 32.0%; caesarean section 22.1%, 22.4%, and 20.1%, in the Flanders, Walloon and Brussels regions, respectively [14,15,16]. The epidural analgesia ratio is 80.0% in Walloon and about 75% in both other regions (15.6%) [14]. Despite the medicalised care, Belgian hospital-based midwives still hold a woman-centred ideology [17]. Nevertheless, maternity care ideology, medicalised or women-centred care, is an attribution of a person, not a profession. The degree of autonomy varies for Belgian midwives; in hospitals, most midwives work under the authority of obstetricians, although this varies amongst hospitals and regions. A midwife in primary care accompanies normal, low-risk pregnant women before, during, and after childbirth. Some primary care midwives autonomously assist homebirths or births in a birthing centre. It may be possible for primary care midwives to assist births in the hospital [18]. The majority of primary care midwives in Belgium, however, only assist women ante- and/or postnatally [11]. In primary care, Belgian midwives work on their own, in group practices, or in public health organisations outside of the hospital [19]. Internationally, midwives employed in primary care settings tend to have more autonomy in the organisation of their work [6,7,20,21].

In Belgium, midwifery education follows a direct-entry program at the bachelor’s level, adhering to the European Directives outlined in Directive 2013/55/EU [22]. As stated in the Directive, Belgian midwives are educated in a full-time programme consisting of at least 4600 h of theory and practice, with at least one-third of the minimum duration based in clinical practice [23]. As a result, students are educated to autonomously provide care for women experiencing uncomplicated pregnancies [24]. Various master’s programmes, such as a Master of Science (MSc) in Nursing and Midwifery, a MSc in Healthcare Management, a MSc in Public Health, or a MSc in Health Education and Health Promotion, are accessible for midwives but not restricted to midwives only [7]. Specialist or advanced roles for midwives are limited to management, research, and educational functions. Advanced roles may include lactation consultants. The concept of postgraduate education (MSc, Doctorate) leading to advanced midwife practitioners is not yet clarified in Belgium [25].

Recently, a consensus definition of midwifery autonomy in Belgium has been developed in order to establish a joint understanding of the concept of midwifery autonomy [26]. The definition comprises critical components related to the work content, professionalism of the midwife, and relationship with others. Together, they encompass the essentials of midwifery autonomy in Belgium. An autonomous midwife is defined as a skilled and experienced health professional who is recognized by society and the medical community. They are capable of working independently, in accordance with their professional profile and the relevant legislation. An autonomous midwife possesses the expertise, authority, and competency to make independent decisions and take control of their work. As an autonomous practitioner, the midwife is responsible and liable for their actions and decisions and is not required to be supervised by other health professionals [26].

Using this research-derived definition, Belgian midwives’ views on their current and future autonomy were explored aiming to strengthen the midwifery profession in Belgium. Belgian midwives generally rated their own professional autonomy as high, but significant differences were observed between hospital-based and primary care midwives and between regions. Midwives working in Walloon felt the least autonomous and recognized of the three Belgian regions (Flanders, Brussels, and Walloon). Midwives with professional experience of more than 30 years and primary care midwives felt less recognised and less respected by other maternity care professionals. A significant majority of participants desired more autonomy in future, but, above all, Belgian midwives wanted to be respected by society and other maternity care professionals. As the literature suggests, midwives’ autonomy may be a subject of debate with other maternity care professionals, women, researchers, and policymakers [27,28].

While the views of Belgian midwives on their current and future professional autonomy was recently studied, the lack of views from other stakeholders in maternity care about midwifery is still a blind spot. The aim of this study, therefore, is to explore multiple key maternity care stakeholders’ views on Belgian midwives’ professional autonomy, in order to shape the future of maternity care.

## 2. Materials and Methods

### 2.1. Design

A qualitative exploratory study using online synchronous heterogeneous focus group interviews.

### 2.2. Participants

The involvement of all stakeholders, and the gathering of their views on a subject, improves the interface between academics and decision makers in health care [27]. In the context of the multidimensional nature of maternity care, we opted for focus group interviews to explore the views of multiple key stakeholders in maternity care [29]. Heterogeneous focus groups with multiple key stakeholders in maternity care were compiled with (1) health professionals, (2) policy advisors, (3) hospital managers, and (4) women’s groups/consumers.

### 2.3. Recruitment

Key stakeholders were targeted using purposive sampling of legally established professional associations, national and regional health related advisory committees, governmental departments, and women’s/consumers associations in Belgium. Our selection of categories was informed by a recent publication (2020) that outlined the organisation of maternity services in Belgium [30]. To ensure comprehensive coverage, we collaborated with 11 content experts: five experts from Flanders (three midwives, one nurse, and one neonatologist) and six experts from Walloon (four midwives, one nurse, and one obstetrician). They gave feedback on a first draft of categories of potential stakeholders. Based on this feedback, the first author adapted the categories. Consequently, the identified stakeholders were contacted by the first author. These content experts were not involved in this study. Table 1 outlines the legally recognised associations representing stakeholders in Belgian maternity care.

We identified 41 stakeholders’ associations in Belgian maternity care: 14 associations representing health professionals, 10 representing policy advisors, and six representing hospital managers, while women’s groups/consumers were represented by 11 associations.

Executive board members of the identified organisations were informed about this study and invited by email to participate three weeks before the planned focus group interviews. Each association was asked to recruit a maximum of two individuals. Potential participants were informed about the study and invited to the focus group. A reminder was sent three days prior to the planned focus groups. An informed consent document signed with wet ink and socio-demographics data (gender, age, association, highest education level, profession, years of professional experience in this profession, language) were collected prior to the focus groups.

### 2.4. Data Collection

In October 2022, focus group interviews were conducted in Dutch and French, as these are the national languages of Belgium, using Microsoft Teams©, Washington, United States. Both languages were used in each focus group. The first and last author formulated four questions to answer the research question. To ensure content validity, two external experts were asked to review the latter questions to increase their content validity (Table 2). We chose to utilize online focus groups because they are a convenient and practical option for busy professionals or individuals with significant time commitments to participate and benefit from [31,32]. To overcome the challenges of remote focus groups and ensure high-quality data, we employed several strategies. These included providing clear instructions for joining the group, setting expectations for participation, limiting the group size to a maximum of 10 individuals to encourage contributions from all, and recording the focus group to avoid any loss of information [33].

During the focus groups, the interviewer (JV) digitally recorded the interview, asked questions, and facilitated the discussion while the observer (MF, an experienced qualitative researcher and lecturer in qualitative research methodology) observed nonverbal clues, ensured that all stakeholders were dealt with in a proportionate way, and that the discussion was not dominated by those with authoritative knowledge. In addition to managing the focus group, detailed field notes were recorded. Focus groups were organised until no new data were forthcoming. All video recordings were securely stored in an onsite locked facility, only accessible to the researchers. The data were not shared or discussed with other colleagues. In order to maintain anonymity, all identifying information was removed. The study is self-funded and ethical approval was obtained from the University Hospital Brussels and the Vrije Universiteit Brussel (VUB) in August 2022 (registration number: B.U.N. 143/202/100/0490).

### 2.5. Data Analysis

Data analysis was performed using thematic analysis [34]. This involved transcribing recordings into text format, familiarization with the data by reading and rereading the transcriptions to gain a thorough understanding of the content, and coding into recurrent and common themes by systematically identifying segments of the data that relate to a specific theme. Throughout the analysis, the researchers (JV, MF) engaged in discussions to ensure the integrity of the analysis and to ensure that the themes were a good fit for the data. Both researchers organised the results, with significant examples of each theme selected and translated into English. To ensure our study was sufficiently transparent, reliable, and reproducible the researchers followed the Consolidated Criteria for Reporting Qualitative Research (COREQ), while designing and conducting the study [35].

## 3. Results

From the 41 invited associations, 24 (58.5%) did participate in this study. A heterogenous group of 27 stakeholders involved in maternity care participated: twelve health professionals, three policy advisors, four hospital managers, and eight service users. Health professionals were represented by obstetricians, paediatricians, general practitioners (GPs), and a nurse. Most participants were aged between 51 and 60 years (n = 9, 33.3%), female (n = 22, 81.5%), with a professional experience of 21–30 years (n = 10, 37.0%), and all educated to a minimal of bachelor level (Table 3: Sociodemographic and professional characteristics of participants).

In total, three focus groups with seven to 11 participants per focus group were undertaken. The focus groups all lasted approximately 120 min (FG1: 120 min, FG2: 121 min, FG 3: 113 min). Each focus group included at least one member from the following groups: health professionals, policy advisors, hospital management, and consumers. In terms of the health professionals, every focus group featured at least one obstetrician and one paediatrician.

### 3.1. Variation between Midwives

The participants in the focus groups reported a wide variation between midwives, with different levels of autonomy related to their professional activities. Commonly, stakeholders viewed hospital-based midwives as quite autonomous despite working under the authority of an obstetrician. While midwifery autonomy was embedded in daily practice in some hospitals, it might be restricted in others. In hospitals with a midwife-led approach and in birth centres, midwives’ professional autonomy was obviously more prominent. Generally, the difference between hospital-based and independent midwives is a divisive issue in the autonomy debate. Some participants focused their attention on independent midwives and homebirths, while others raised concerns about the competencies of independent midwives regarding risk selection and referral to other health professionals. Conversely, a lack of competence in risk selection was not mentioned with regard to hospital-based midwives.

Variation in practices of primary care midwives was reported. While primary care midwives were perceived as autonomous, many differences and grey areas between them were expressed by stakeholders. This was highlighted by a health professional as such:


*“So I think …., that to summarise there is a huge heterogeneity [in primary care midwives], and that there is a need for quality of care. I don’t mean that not everyone is doing their best to give quality but that, indeed, sometimes that goes in different directions”.*
(FG 3_Health Professional 2)

The analysis of the data resulting from the focus group interviews resulted in five themes (Figure 1): (1) The autonomous midwife is adequately educated and committed to continuous professional further education, (2) The autonomous midwife is competent, (3) The autonomous midwife is experienced, (4) The autonomous midwife assures safe and qualitative care, and (5) The autonomous midwife collaborates with all stakeholders in maternity care. Each of these themes will be explored in more detail below. Excerpts from relevant quotes from the focus groups are provided as illustrations.

The autonomous midwife is adequately educated and committed to continuous professional further education.

Several stakeholders identified education, collaboration, and competence as key pillars for autonomous midwifery. Most stakeholders believed that education and continuous professional education are prerequisites for midwifery autonomy. While midwifery education currently equips midwives to work in the hospital, some stakeholders argued that newly graduated midwives lacked competence for primary care, especially for performing homebirths. To increase competence in primary care, more practical placements and clinical experience are suggested by some stakeholders. This was made explicit by one of the stakeholders:


*“I think that self-employed midwives who actually do so [performing homebirths] are insufficiently trained and experienced. I think, … that a midwife would act better if she had worked, an extra year for example, 1 or 2 years in a obstetric unit with at least 100 births a year, I say something. Because I think that you are then so much better skilled to make appropriate judgments at home”.*
(FG 1_Health Professional 1)

Some stakeholders suggested including additional topics in undergraduate midwifery education, and early risk detection and referral were the most mentioned. To facilitate communication amongst maternity care professionals, some stakeholders suggested organising interprofessional communication courses in undergraduate and continuous professional development education.

As there were increasing demands for newly graduated midwives, most stakeholders agreed that it is challenging to include all necessary study contents in a three-year midwifery programme. However, the majority of stakeholders did not clearly indicate how midwifery education needs to be restructured. The completion of an in-depth study or specialisation year of one or two years for midwives who would want to work independently was obvious for some stakeholders though. One stakeholder referred to a recent initiative of a postgraduate program in the Walloon region of 30 European Credit Transfer and Accumulation System (ECTS) aiming at the acquisition of advanced midwifery competences in uncomplicated birth:


*“We also question education. This is why recently at [two universities in the Brussels Capital Region] a further training in advanced midwifery practice was established. We know that midwifery education prepare midwives to work in the hospital under supervision, … and that they are not always prepared to work autonomous …”.*
(FG2_Consumer 1)

As the evidence evolves, stakeholders expect midwives equally to evolve and improve their competences continuously. Interprofessional meetings and compulsory training to develop specific competences were viewed as necessary for all midwives. Some doctors and hospital managers clearly preferred to organise their own professional development activities for midwives. These further training activities are perceived as quality education and in line with the demands of daily practice. Those stakeholders recommended that primary care midwives participate in those activities, so their practices will be aligned with hospital policies. Additionally, this provides the midwives an opportunity to connect with the maternity care team. These doctors and hospital managers were convinced that the established interprofessional connection enhances confidence and facilitates collaboration.

All stakeholders believed that all further professional training courses should be tailored to up-to-date scientific knowledge. Moreover, it was suggested that official accreditation of professional development courses would benefit midwives’ professionalisation. Since there are no formally supervisory measures for continuing professional education in midwifery yet, certain stakeholders recommend that a regulatory authority oversees the continuous professional development activities of midwives. Additionally, reference was made to the recent Belgian Healthcare Quality Law (Kwaliteitswet Gezondheidszorg, 2022). This law requires Belgian health professionals, including midwives, to maintain a portfolio that demonstrates their skills and experience. Nevertheless, some stakeholders considered that this law is a good initiative, but yet too vague while lacking implementing decisions.

### 3.2. The Autonomous Midwife Is Competent

The theme ‘The autonomous midwife is competent’ highlights the importance of midwives’ competence from a stakeholder perspective, which includes having excellent knowledge, being alert to risks, and collaborating with other professionals. While most stakeholders agreed that midwives are competent, some expressed doubts about their competences in certain areas and suggested clarifying midwives’ competences.

Competence was identified as a major attribute of a midwife, and this was a matter of course for all stakeholders. All stakeholders acknowledged that most midwives are competent and adequately educated for their tasks, regardless of her professional setting. Competence was viewed by stakeholders as holding an excellent knowledge of physiology and pathology, being alert to risks and warnings, proactive behaviour, and collaboration irrespective of the scope of practice. A stakeholder expressed this as follows:


*“She [the midwife] has completed the necessary education so that she can call in the required help when needed. This need to be in her basic training irrespective if she works at home, at a birth centre or in a hospital. She needs to be competent to identify when and where to refer”.*
(FG2_Consumer 2)

All stakeholders acknowledged that midwives may only act within the legally defined framework of their profession. This requires that midwives know their exact boundaries and adhere to them. Midwives need to make proper assessments of each situation, make appropriate referrals, and engage with other health professionals in a timely manner. It is expected that a midwife is never too confident and stays alert all the time. Some stakeholders mentioned that this is challenging in reality, as a situation can quickly deteriorate while the distinction between pathology and physiology is not unambiguous.

Midwives are most autonomous while working in antenatal and postpartum care. Some stakeholders believed that midwives should be allowed to prescribe more medication in pregnancy. Conversely, several stakeholders wished that some of the boundaries of midwives’ competences be reassessed. These boundaries referred to the follow-up period of babies by the midwife. Those stakeholders were doubtful about midwives’ competences in the follow up of a baby until three years of age, giving examples of advice about infant vaccination or toilet training.


*“I think, for example, giving advice on potty training, is that still a task of an independent midwife? To what extent, … because you can stretch it to … 25 years, to what extent does midwives’ competence reach? I believe that the professional profile and legislation should be guiding in this”.*
(FG2_Health Professional 2)

Midwives’ competences are not well known by society and other maternity care professionals. Some service users did not know exactly what midwives can do and their limits, they suggested clarifying midwives’ competences to women and other occupational groups. Overall, maternity care professionals trust the midwife if they consider her competent. When health professionals know each other’s competences it promotes mutual trust. Likewise, trust in each other’s competence facilitates early referral in case of doubt, which was highlighted by one of the stakeholders:


*“When I may speak for the users, I think the expectations are that they [midwives] are competent, that they are capable, that they can detect if there are problems, if there are pathologies… then refer to other professionals when this is the case”.*
(FG1_Consumer 5)

Clear communication with parents about the midwife’s experience is needed, as well as competences and the interprofessional perinatal network she is part of. This communication needs to be done with respect to other maternity care professionals’ competences and their respective roles in maternity care. Conversely, some stakeholders argued that the needs of women might be unknown to the health professionals:


*“So, the needs of the patients are unknown to us,… and it is our role as professionals to talk about their options, to help them make informed choices … And I think it is our duty as health professionals to inform them correctly”.*
(FG3_Hospital management 13)

### 3.3. The Autonomous Midwife Is Experienced

Through experience, intuition is developed which is important to assess labour progression and the early detection of abnormalities. Experience was identified as key to mastering midwifery activities, which was pointed out by one stakeholder:


*“In your definition [definition of midwifery autonomy in Belgium] there is a word that drew my attention, and that is ‘mastery’. And there I support [a stakeholder from hospital management,] to achieve mastery you must work with different professionals. To be autonomous it is the mastery, the anticipation and the continuity of care”.*
(FG3_Health Professional 4)

From the perspective of the stakeholders, independent midwifery should only be allowed if the midwife has extensive clinical experience and is supervised by a senior colleague at the start of her career. A health professional stakeholder made reference to similarities in medical education, where doctor specialists in training are educated under the supervision of an experienced mentor. These mentors need to have educational and clinical expertise. Newly graduated midwives do not have enough experience and are not sufficiently equipped to work independently. Conversely, a stakeholder argued that less experienced midwives’ judgement is not accurate in recognizing problems in a timely manner:


*“I think you need a minimum experience in performing births in a hospital to know em, … when it suddenly unexpectedly goes wrong. Because if you don’t have real work experience in the hospital, I think you’re missing some competences … So a minimal experience, … really working as a midwife and in the hospital, I find that indispensable as a paediatrician”.*
(FG2_Health Professional 2)

Some stakeholders doubted the interprofessional collaboration competences of less-experienced midwives. Particularly, new midwives need to finetune their practice within a multidisciplinary team. Clinical experience and collaboration with other health professionals is less a matter of concern for hospital-based midwives as they can rely on a multidisciplinary team at all times.

### 3.4. The Autonomous Midwife Assures Safe and Qualitative Care

Most stakeholders stressed that safe and quality care should be a priority at all times. Stakeholders noted that each health professional can work autonomously but never at the expense of mother or child’s safety. Some stakeholders were of the opinion that midwives work independently but in a hospital only. They stressed that midwives are competent to lead uncomplicated labour and childbirth, but only in a setting with a multidisciplinary team as back up. Under that condition, stakeholders were in favour of more autonomy for hospital-based midwives. A stakeholder expressed this as follows:


*“For me, autonomy of midwives in Belgium should be limited to a physiological pregnancy, … but always in a medical setting,… So, autonomy of midwives is certainly possible, it is done in many hospitals where midwives can consult autonomously, but in collaboration with a doctor who can always give their opinion. Like in the labour room, midwives are equipped for it to be, perfectly possible that the midwife performs normal births, I am the first defender of that, but always in a medical setting”.*
(FG2_Policy advisor 1)

Giving birth outside the hospital was seen as taking unnecessary and avoidable risks by some stakeholders. In the opinion of others, the hospital may induce a false sense of safety, resulting in unnecessary medical interventions. Some argued that in a hospital the emphasis is less on one-to-one or women-centred care. Giving birth in the hospital does not necessarily mean that all health professionals present are experienced and competent, as argued by a stakeholder:


*“What secures birth is the human and not the machines, it’s not the hospital that secures birth. It is the caregivers who are well equipped, well trained and work together. Autonomy is about collaboration, but with respect to the expertise and knowledge of each one. And so it’s not because you are a doctor that you are a good doctor, …, it’s not because you are a midwife that you are poorly trained, that you have no experience and that you don’t know your limits”.*
(FG2_Consumer 1)

Some stakeholders advocated for the use of a checklist in pregnancy with clear agreements with other maternity care professionals to anticipate for the unexpected and to safeguard safe care. These stakeholders found that the role of the different maternity care professionals and hospitals need to be outlined. These arrangements need to be drawn up with the different health professionals and regulate the cooperation in the multidisciplinary team. A stakeholder expressed this as such:


*“A pregnancy is a period when a kind of checklist needs to be used to anticipate and to make clear agreements between your network, between paediatricians, with obstetricians …. So now, I think that is a very important issue because we are actually talking about, a care path and about agreements and about em limits”.*
(FG3_Health Professional 3)

Most stakeholders were convinced that the use of protocols, guidelines, and care pathways reduce risks. To assure safe and qualitative care, maternity care needs to be structured and standardized in their opinion. Rules on health care professions must be clearly defined by their respective established professional associations. The role of the professional organisations is to provide quality guidelines to their members. Therefore, some stakeholders suggested that the membership of midwives of an established professional association should be mandatory. That not all midwives are members of an established midwifery professional association may impede quality of care according to some stakeholders. Evidence-based care is supposed to be embedded in midwives’ practice. Moreover, some stakeholders thought that the compliance with evidence-based guidelines should be actively promoted by midwifery professional associations. Some stakeholders, particularly health professionals, argue that the work of independent midwives sometimes lacks scientific substantiation. Additionally, they found it striking that some of these midwives are recommending non-conventional therapies such as laser removal of the lingual frenulum, food supplements, and osteopathy for newborns, e.g.,:


*“And what I find difficult is that they [independent midwives] are a group is that often goes quite the alternative tour and yes, … I have seen few children who have not been sent to the osteopath, recommended to take supplements—that cost a lot of money—but which isn’t much of proven value”.*
(FG2_Health Professional 2)

One stakeholder advocated for more midwifery regulation in Belgium. This expert advocated for a hotline where midwives and other maternity care professionals involved in doubtful practices can be reported. This initiative is suggested to be tailored to the way in which the Order of Doctors (Orde der Artsen) deals with professional ethics, advice, and disciplinary powers for Belgian doctors.

### 3.5. The Autonomous Midwife Collaborates with All Stakeholders in Maternity Care

Autonomy might be misunderstood as working independently and in isolation. This was pointed out by a stakeholder as follows:


*“Autonomy, that sounds like ‘I work on my own, and this is my field of expertise and you must stay away’, … and if we would collaborate and respect each other’s competences …, with that we would move forward”.*
(FG3_Policy Advisor 1)

Some stakeholders mentioned that it is a danger if midwives do not collaborate with other health professionals. A low threshold between health professionals is seen as important, as it facilitates communication, early referral, and safe care. Independent midwives need to elaborate an interprofessional network in stakeholders’ views. Regular debriefings with other maternity care professionals should be structurally embedded as a form of critical peer review, which would increase confidence and mutual trust. Smooth and easy lines of communication between health professionals should be facilitated, according to several stakeholders.

One stakeholder experienced a mistrust between obstetricians and independent midwives noting that obstetricians often think that independent midwives go too far in their physiological approach and misinform women about maternity care in the hospital. Stakeholders called for an opening up of the discussions between all maternity care professional to restore respect and confidence:


*“Nowadays obstetricians are absolutely willing to respect physiology as much as possible. But we know that many independent midwives do not trust, … So I regret that, I think it really, really [accentuated] is time that obstetricians and independent midwives come back together, because I think we do not really have so many differences in vision at the end, … together we can make good progress”.*
(FG1_Health Professional 1)

Stakeholders were convinced that if maternity care professionals keep women’s interests as a focus, it will facilitate smooth and respectful interprofessional collaboration. Health professionals should have to resume a debate and this debate needs to be respectful, while refraining from expressing opposing views publicly and showing respect to each other. A stakeholder expressed this as such:


*“… to act in a confraternal way and, … we must restrain debate, … As obstetricians we have to stop saying ‘yes, but the midwife knows nothing’ and the midwife has to stop saying ‘yes, but the obstetricians are always exaggerating’. When we have that mutual reserve, it is obvious that we will move on much better”.*
(FG3_Health Professional 4)

Maternity care professionals need to know and respect each other’s competences, which is fundamental for interprofessional collaboration. All maternity care professionals must strive for respectful interprofessional communication and collaboration. Respectful interprofessional collaboration will enhance trust and add to professionalisation. Competent midwives will be respected by other maternity care professionals, leading to mutual trust and respect. Most stakeholders acknowledged midwives as trusted and equal members of the maternity care team with common goals. A stakeholder expressed her vision on interprofessional collaboration:


*“I can testify that it [respectful collaboration] goes very well with the paediatricians, obstetricians, … All the health professionals surrounding the midwife, where everyone has a place and respects each other in what they do. When it brings value to the patient, it is good for the patient and the continuity of care and most important …, I think, for each one of us”.*
(FG3_Hospital Management 1)

## 4. Discussion

In this study, we aimed to capture the views of stakeholders in maternity care on Belgian midwives’ professional autonomy. Stakeholders exposed many differences between Belgian midwives’ levels of professional autonomy related to their professional domain. Our research identified competence, education, collaboration, and respect as prerequisites for midwives’ professional autonomy. These prerequisites are essential for improving the quality of midwifery care and ensuring the safety and well-being of women and newborns and should be considered in the context of the identified themes.

### 4.1. Competence

All stakeholders in maternity care acknowledge that competence is a major attribute for quality care. Midwives can develop competence by gaining experience in all domains and settings [36]. Most stakeholders find that the use of guidelines ensures quality of care. Evidence-based guidelines need to be developed in collaboration between obstetricians and midwives [37]. Women should be involved in the development of the guidelines as the health professionals’ compliance with the guidelines might be considered more important than women’s wishes [38]. However, while Belgian midwives are convinced of the importance of evidence-based practice guidelines, they do not believe that guidelines enabled them to provide woman-centred care [37].

Differences in competences among hospital-based and independent midwives were expressed. Some participants questioned the competences of independent midwives regarding risk selection and referral to other health professionals. This lack of competence in risk selection was not mentioned in regard to hospital-based midwives. Belgian doctors, along with hospital-based midwives, have negative views on homebirths [17]. When they meet a woman in hospital who intended to have a home birth, she was mostly referred due to complications. As hospital birth is what hospital-based midwives know, this is probably what they consider the safest option [17]. While hospital-based midwives may view homebirth as an intrapartum risk [39], this perspective may not be shared by midwives working in hospitals where homebirths are offered as an alternative maternity care option for women, as is the case in the UK and Australia, among other countries [40].

Belgian midwives who perform births independently, however, must comply with legal requirements [41]. The aim of this regulation is to provide safe and quality care. Midwives who have recently graduated may only assist births under the supervision of a senior midwife. The advice is adopted by insurance companies and is binding for independent midwives. Additionally, in 2016, the Flemish Professional Association of Midwives introduced the ‘Good Practice Logo’ (GPL) for independent midwives, a quality label for midwifery practice. To comply with the GPL, midwives have to meet several prerequisites such as evidence-based care and being a member of a professional association and a midwifery network in Flanders or Brussels [7]. Compliance with the GPL conditions is assessed annually by random sampling by the association, despite compliance not being a legal obligation. Additionally, the professional association has no regulatory or supervisory powers. Some stakeholders suggest adopting the model of the Order of Doctors. A compulsory registration in the Order of Doctors applies to all doctors wishing to practice in Belgium. The Order ensures the moral integrity and professional autonomy of the profession and the confidence of society in the doctor. In this respect, evidence-based practice remains the unconditional criterion for Belgian doctors [42]. Nevertheless, individual health professionals may be influenced by a variety of factors that can lead them to deviate from evidence-based practice, such as personal beliefs, financial incentives, and social pressures [43].

### 4.2. Education

While Belgian midwifery education adheres to the European Directive and the International Confederation of Midwives (ICM) Global Standards for Midwifery Education [7], some stakeholders have recommended incorporating supplementary topics into undergraduate midwifery education. Suggestions were made to include more interprofessional education and collaboration in both undergraduate and postgraduate midwifery education. Interprofessional education and collaboration are linked [44] and are acknowledged as improving students’ performance in health care [45]. Students from different disciplines learn about, from, and with each other in a safe learning environment [46]. The involvement of medical students and trainees in obstetrics need to be considered to promote interprofessional competences [47].

While a majority of stakeholders do not clearly indicate how exactly midwifery education would need adjustment, stakeholders particularly believe in the idea of lifelong learning. Additionally, they expect that midwives continuously educate themselves professionally. Some recommend a supervisory authority for professional development activities of midwives. Most stakeholders are convinced that all continuous professional development courses should be tailored to up-to-date scientific knowledge.

### 4.3. Collaboration

Most stakeholders viewed midwives as equal members of the maternity care team. Maintaining safe, quality, and respectful care as the common focus facilitated respectful interprofessional collaboration. Maternity care professionals needed to be aware of each other’s competences and respect them, which was crucial to interprofessional collaboration. Therefore, a transparent care path and collaboration between maternity care professionals is desired [48], and within this collaboration, the autonomy of all maternity care professionals need to be respected [44]. Finding the balance between a high level of professional autonomy amongst maternity care professionals and good collaboration is, however, a challenge [49].

Midwifery associations were viewed to be key to the integration of the profession in health systems and for holding the profession together. A strong association is the foundation to create quality midwifery systems and support midwifery regulation and accreditation [50]. Professional associations should push for collaboration in regional maternity collaborative networks enabling all maternity care professionals to join the network meetings. Municipalities and associations of maternity care professionals should work together to get the respective professionals and consumers in touch with each other [48]. This interprofessional collaboration should be structurally embedded in maternity care and promoted by professional associations. In the Netherlands, health insurance companies have made collaboration networks a requirement during their negotiations with midwives and hospitals for agreements, as of 2022 [51].

Safe care encompasses the involvement of women and a relational model of care within a collaborative and evidence-based health system [52]. To enable fully informed consent, midwives should share birth information, in all its forms. To be with the woman, they should be open to the woman’s choices [53]. Birth plans that facilitate shared decision-making and women’s sense of autonomy and control before, during, and after giving birth is critical. When discussing the birth plan, exploring different scenarios may help women prepare for unforeseen circumstances [54].

### 4.4. Respect

Relational models of care such as midwife-led continuity of care, which is cost effective and guarantees optimal outcomes, are currently ignored [52]. The added value of midwife-led continuity of care is unknown to Belgian policy makers [55]. The first freestanding midwifery-led unit within a Belgian hospital, installed in 2014, received a lot of attention from stakeholders in maternity care, including policy makers from across the country. All initiatives, however, appear to be concerned about the lack of government funding for this kind of service. Belgian midwives and consumers have voiced the need for the introduction of midwife-led continuity care [56].

If the identified prerequisites of midwifery autonomy are adopted at all levels of the maternity care system and by maternity care professionals, most women will have a safe outcome, related to a positive birth and motherhood experience. When further outlining the future of midwifery in Belgium, which include midwifery association and education, it is essential to consider the results of this study. Stakeholders emphasized that listening to women’s voices to shape the future of maternity care is needed. Midwives collaborating with the woman, as with all maternity care professionals, will be critical for future Belgian maternity care [44]. Midwife-led continuity of care needs academics, policymakers, and governors in established professional midwifery associations to put this relational model of care up for debate with healthcare politicians, health services, insurances, and consumers on a macro level [57].

### 4.5. Strengths and Limitations

#### 4.5.1. Strengths

One of the strengths of this study is that it is part of a comprehensive research project that aims to explore the professional autonomy of Belgian midwives. The study gathered insights from midwives, stakeholders, and midwifery students, to gain a well-rounded understanding of the topic. To explore maternity care stakeholders’ views, we performed online synchronous heterogeneous focus group interviews. Participants interacted with the interviewer and each other live. Online focus groups offer convenience, reduced costs, access to a wider range of participants, and reduced social desirability bias. However, they also have limitations such as technical issues, potential for distractions, and limited interaction between participants [32]. We achieved a professional heterogeneous composition regarding professional background, education and professional experience [34]. Nevertheless, all our participants have professional experience and expertise in maternity care in common [33]. We invited stakeholders through their respective professional associations. The inclusion of stakeholders from established professional associations increases the likelihood that they are committed to the subject and can represent their (professional) group [27]. Nevertheless, at the beginning of each focus group, the interviewer stressed that while each stakeholder participated as a representative of their association, they were not necessarily expected to echo their associations’ views. Moreover, they were mainly invited because of their expertise in Belgian maternity care.

#### 4.5.2. Limitations

Seventeen associations did not participate in this study (41.5%), which curtails its generalizability. The major reason for non-participation was that the subject of midwifery autonomy did not fit with the associations’ expertise (mentioned by one health professional, four policy advisors, one hospital management, and two consumers associations). Additionally, two associations reported lack of time (one health professional and one consumers association), while one policy advisors’ association stressed that they wanted to maintain a neutral attitude towards both obstetricians and midwives. Nevertheless, an adequate participation of stakeholders in each focus group was achieved. Moreover, in each focus group, at least one member from health professionals, policy advisors, hospital management, and consumers participated. Additionally, each focus group consisted of a minimum of one obstetrician and one paediatrician. Extensive professional experience amongst the stakeholders was observed in all focus groups; 22 participants (81.5%) had a professional experience of more than 11 years.

We did not critically consider potential relationships between stakeholders and researchers, which could include issues of power dynamics, ethical considerations, and the potential impact on the findings. To address this issue, participants were invited to provide feedback on the findings [58]. A short summary of the findings was forwarded for feedback two months after the discussions. Only one amendment was received, suggesting that aspects of midwives’ liability should be more emphasised. As the aspect of liability had only been cited in one focus group, the first and last author opted not to elaborate on this aspect.

## 5. Conclusions

This is the first study to explore Belgian midwives’ autonomy through the lens of maternity care stakeholders. The involvement of key maternity care stakeholders in focus group interviews was critical to shape the future of maternity care. Municipalities and established professional associations need to collaborate and unite maternity care professionals. The structural embedding of women in such an organisational structure will be critical to shaping the future of maternity care in Belgium. In the proposed maternity collaborative framework, all maternity care professionals need to be aware of each other’s competences, while respecting each other’s autonomy. The foundation for successful interprofessional collaboration must be established through interprofessional education for healthcare professionals.

While most midwives are perceived as trusted and equal members of maternity care teams, the difference between hospital-based and independent midwives is divisive in the autonomy debate. A compulsory registration in an Order, a regulatory body with supervisory powers, may reduce undesirable differences among Belgian midwives. It is emphasised that adopting the model of an Order will help to ensure safe and quality care and catalyse midwives’ autonomy and professionalisation.

## Figures and Tables

**Figure 1 healthcare-11-01231-f001:**
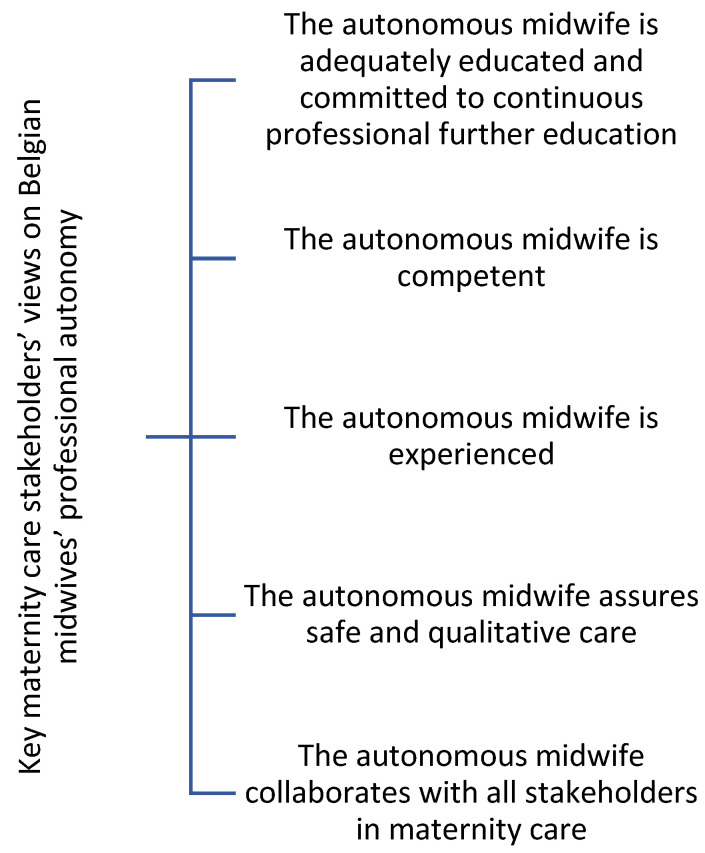
Identified themes.

**Table 1 healthcare-11-01231-t001:** Identified legally established associations representing stakeholders in Belgian maternity care.

Health Professionals	Policy Advisors	Hospital Management	Consumers
Flemish Association for Obstetrics and Gynaecology	Planification commission	Council of University Hospitals Belgium	Flemish patient platform
Professional association of Belgian obstetricians and gynaecologists	Federal Knowledge Centre for Healthcare KCE	Board of nursing managers NVKVV Network Nursing	Representatives of patients in the Federal Knowledge Centre for Healthcare KCE
Royal College of Gynaecologists Obstetricians of French Language of Belgium RVB	Study centre for perinatal epidemiology, Flanders	Flemish Association for Nursing executives	Women’s Council FERM
Belgian Group of French-speaking Paediatricians	Perinatal Epidemiology Center, Brussels and Walloon	Flemish Hospital network	Health Services Users League
Flemish Society for Paediatrics	Federal Council of Midwives	Belgian Association of Nurses and National Federation of Nurses of Belgium: Board of Directors of Nursing Departments	Platform for a respected birth
Belgian Society for Paediatrics	National Institute for Health and Disability Insurance	Federal Council of Hospitals	Together for respectful birth
Professional association of General Practitioners in Flanders and Brussels	Federal Public Service Health, Food Chain Safety and Environment		The world according to women
Belgian Group of General Practitioners	Federal Public Service Social Security		Dutch-speaking women’s Council
College of physicians for the mother and the newborn	Zorgnet ICURO, umbrella organisation of the Flemish general hospitals, initiatives in mental healthcare and social profit facilities in geriatric care		Council of French-speaking women of Belgium
Belgian Society for Neonatology	Flemish health ambassador		Feminine Life
Child and Family Services			Flemish association for parents of incubator babies
the Office of Birth and Childhood ONE			
Federation of Francophone Medical Centres and Health Collectives			
Scientific Society of General Medicine			

**Table 2 healthcare-11-01231-t002:** Interview guide.

**Engagement Question**
How do you perceive Belgian midwives’ autonomy in everyday practice?
**Exploring questions**
What are your expectations of midwives working as autonomous practitioners?
To what extent do you think that midwives should act autonomously?
What would you think are factors/stakeholders that influence midwives’ autonomy?
**Probes (in order to minimise misunderstandings)**
Can you please tell more about this?
Please, help us understand what you exactly mean by that?
Can you give us an example of that?
**Exit questions**
Is there anything additional you would like to say about midwifery autonomy?
Of all things discussed today, what do you think is the most important?

**Table 3 healthcare-11-01231-t003:** Sociodemographic and professional characteristics of participants.

		Health Professionals n = 12	Policy Advisors n = 3	Hospital Management n = 4	Consumers n = 8
Gender (female, male)	Female	9	2	3	8
	Male	3	1	1	0
Age (years)	20–30	0	0	0	1
	31–40	1	0	1	4
	41–50	3	2	0	0
	51–60	6	1	1	1
	>60	2	0	2	2
Native language (Dutch, French)	Dutch	7	2	2	4
	French	5	1	2	4
Education level (highest completed education)	No education/Primary education only	0	0	0	0
	Secondary education	0	0	0	0
	Tertiary education	12	3	4	8
Professional experience (years)	<5	0	0	0	3
	5–10	0	0	0	2
	11–20	3	1	1	2
	21–30	7	1	1	1
	>30	2	1	2	0

## Data Availability

The data presented in this study are available on request from the corresponding author.

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
