# Peer review of "Key Maternity Care Stakeholders’ Views on Midwives’ Professional Autonomy"

_healthcare, 2023, doi:10.3390/healthcare11091231_

Round 1
Reviewer 1 Report
Interesting article. It reports the specific situation of Belgium, a country that integrates the European Union where the training and professional practice of midwives is defined in European Directives, however neither the 2005 nor the 2013 are mobilized, which leads to question some aspects presented at the level of training essentially. The study would benefit from this integration in the discussion of results.
Methodologically it is adequate and well reported.
Still at the level of discussion, prerequisites for the professional autonomy of midwives are found, which, while being perceptible, move away from the identified emerging themes. It would benefit from a better explanation of this option and organization in the discussion.
Strengths and limitations and appropriate conclusions.
Around 61% of references integrate a time frame in the last 5 years, which is positive.
Author Response
Dear reviewer,
Please see the attachment.
Kind regards on behalf of the authors.

Reviewer 2 Report
This is an interesting paper which highlights the apparent challenges faced by the profession as a consequence of the lack of understanding of what the profession actually does. It is good to see the stakeholder's perspective being considered in this regard as at the end of the day it is the stakeholders who make the determination that the care provided is culturally safe and as such this impacts on the professionalism aspects of the profession.
Design indicates that focus group interviews were conducted, with this being again stated in line 123 and 125, however at line 118 there is reference to "semi-structured interviews". There is a difference between the two methods and there needs to be consistency in which was used. If focus groups were used this needs to be the method described, if both were used this needs to be clearly outlined.
Good integration of the quotes from the participants and this adds to the discussion. The identified themes are clear and well supported with the use of evidence from the participants.
Good discussion which is well supported with the use of appropriate literature.
Consideration of the strengths and weaknesses of the study is sound.
Line 38 - 39 needs amendment to be grammatically correct.
Line 108 - consider rewording full sentence on this line so it reads more fluently
Line 149 - 150 edit needed to flow fluently
Line 161 - 162 - address the repetitiveness of the beginning of the sentences
Author Response

(The authors gave the same response as above.)

Reviewer 3 Report
Very interesting study. I think the quality of the paper would be improved if you could make some additional corrections based on the comments below.
Participants
To be able to determine the appropriateness of the subject for this study, you mention that you used purposive sampling, but please provide additional rationale for selecting these four categories.
Data collection
Interviews are conducted remotely rather than face-to-face. There are some difficulties when discussing with several people, such as the fact that technically they cannot speak at the same time. It would be better to describe how these challenges were addressed and how the quality of the data was ensured.
Data analysis
Please add the details of thematic analysis.
To ensure our study was sufficiently transparent, reliable and reproducible, please describe the specific innovations you are making
Discussion
I think it is a very interesting study. Was there a difference in the nature of the themes that emerged from the subjects of this study and the autonomy that midwives alone consider? It would be good to add to the strengths of this study.
Author Response

(The authors gave the same response as above.)

Reviewer 4 Report
Thank you for asking me to review this interesting paper which aimed to explore midwives' professional autonomy from the perspective of key maternity care stakeholders. This appears to be a novel area to explore and provides valuable insight which the authors indicate can be used to inform and establish further professional regulation in Belgium.
The introduction provides the reader with context and clearly identifies a gap in the evidence base and as such justifies the aim of the study, which the authors have clearly stated. References appear to be appropriate for the subject area (although this is not my area of expertise).
It is my opinion that the Materials and Methods section of the paper could be strengthened. Recruitment of participants is clearly provided and justification is given for the approach used. Justification for decision to use online focus group interviews would be of benefit here. It is not apparent what language was used in the focus groups and if not English, then the need to translate the transcripts. Given the mix of participants in these groups (professionals, managers, policy makers and consumers), the paper may consider how the potential for discussions to be dominated by those with authoritative knowledge was mitigated by the researchers. Importantly critical examination of the relationship between participants and researcher has not been considered.
Some key ethical issues have been described - including consent, protecting anonymity, data protection and that the study received appropriate scrutiny through ethical review.
There are some inconsistencies in terms used under data collection reference to semi structured interviews (line 118) and under data analysis the authors refer to conducting a survey (line 138). Further clarity here may be useful.
The results presented include some use of quotes to illustrate points made, although inclusion of more data excerpts would strengthen this section. At times, the narrative around the results is a little confusing as it appears to be mixing discussion with findings. This may be down to writing style and language used, but I advise strengthening this section so that it is clear that it is being driven by the data rather than the opinion of the authors. This issue is further exacerbated by the frequent change of tenses. When describing the findings, it would be preferable to use past tense (not present tense).
The conclusions appear to be consistent with the findings. However further discussion of strengths and limitations around methods (especially strengths and limitations of online focus groups) would be of benefit.
Author Response

(The authors gave the same response as above.)

Reviewer 5 Report
Thank you for the opportunity to review this paper.
Noteworthy topic but how this was undertaken I am not so sure about.
Main issue is that there is a lack of explanation about many things. Assume the reader is from Belgium and knows how maternity care is provided there.
Might be helpful to define what is meant by autonomous midwife and at what stage in their career would they considered to be this or is the thinking this should be the case from graduation.
line 47 'in Flanders' needs to be outside of the bracket as part of sentence
line 56 not clear regarding other countries and why put Belgian midwives - surely in other countries there are midwives from that country that practice
line 71 is about a study undertaken in Netherlands. The previous paragraph is about the definition from Belgium. Need to introduce this study otherwise it reads like it is about the definition. Also needs an introduction/explanation as to why this is being used when discussing midwives in Belgium = context.
Would be useful to define who the health professionals are and the health managers = what discipline they came from. It is not clear why these stakeholders were chosen/justified as I would question the ability of many of these to be able to know if midwives are autonomous and why. As well as the fact that some would be bias either towards or against midwives autonomous role. Line 264 supports my concerns - they do not even know what midwives do so how can they know if midwives are autonomous or not
May also be useful to provide the context of how midwives practice in Belgium as it would help explain.
Not clear what the role of the eleven content experts was. Recruitment section is not clear and may benefit from rearranging.
How did you develop and content validity the questions.
Why interviews in Dutch and French. Was this study undertaken outside of Belgium about Belgium midwives or what.
Role of observer is more the what is written in line 124 - observe body language and record filed notes plus to manage the focus group.
How analysis was undertaken should be referenced.
line 141 - 12 should be twelve as start of sentence
need to refer to the tables in the test and say what they contain
line 154 is this the first theme - introduce this. Not clear what the section is.
what is a primary care midwife
how are midwives educated - are they all undergraduate education. It is just that line 185 does not make sense. What is the relationship between these points.
line 188 - competence in which domain
line 196 explain what is meant by additional topics - not really identified the issues here. Skimmed over them. Explain postgraduate program and where this fits. How much clinical is undertaken in the undergraduate program and when graduated do midwives have a special new graduate supportive program/transition/internship or what.
line 227 I assume from this that there is no requirement for ongoing registration for the undertaking of so many hours of continuing professional education. Need to explain.
line 241 wonder if this is more about scope of practice
This theme needs rethinking as it is poorly constructed
Line 287 - to achieve mastery you must work with different professionals - not clear how this is related to competence and experience.
line 294 - should be 'do not'
line 303 concerned about the logic with this point
line 355 I know this is what the participants said but there are other health professionals (doctors for one) whose work is not scientifically substantiated either - need to be very careful about making such statements. Again questions the appropriateness of the stakeholders to be commenting on midwives being autonomous or not
line 369 this theme is more about collaboration with women or being women centred. This is a different issue to being autonomous as it is the role of the midwife to be with women and women centred. Collaboration with health professionals is something completely different and something that is discuss under theme 3. This theme is very convoluted and not clearly distinguished as there are more then one theme within this theme.
line 441 incomplete sentence
line 451 this is not always the case in other countries. There are some maternity units in UK and Australia (among others) where homebirths are part of the hospital maternity care as another option for the women
line 537 - not clear if this first paragraph is discussing strengths or limitations. Same for whole section.
line 386 'do not'
Author Response

(The authors gave the same response as above.)

Round 2
Reviewer 5 Report
comments not adequately addressed
